# Plaintext-Related Dynamic Key Chaotic Image Encryption Algorithm

**DOI:** 10.3390/e23091159

**Published:** 2021-09-02

**Authors:** Zeming Wu, Ping Pan, Chunyang Sun, Bing Zhao

**Affiliations:** Electronic Engineering College, Heilongjiang University, Harbin 150080, China; 2201669@s.hlju.edu.cn (Z.W.); 2201647@s.hlju.edu.cn (P.P.); 2191309@s.hlju.edu.cn (C.S.)

**Keywords:** image encryption, plaintext-related, dynamic keys, chaotic systems

## Abstract

To address the problems of the high complexity and low security of the existing image encryption algorithms, this paper proposes a dynamic key chaotic image encryption algorithm with low complexity and high security associated with plaintext. Firstly, the RGB components of the color image are read, and the RGB components are normalized to obtain the key that is closely related to the plaintext, and then the Arnold transform is used to stretch and fold the RGB components of the color image to change the position of the pixel points in space, so as to destroy the correlation between the adjacent pixel points of the image. Next, the generated sequences are independently encrypted with the Arnold-transformed RGB matrix. Finally, the three encrypted images are combined to obtain the final encrypted image. Since the key acquisition of this encryption algorithm is related to the plaintext, it is possible to achieve one key per image, so the key acquisition is dynamic. This encryption algorithm introduces chaotic mapping, so that the key space size is $10^{180}$. The key acquisition is closely related to the plaintext, which makes the ciphertext more random and resistant to differential attacks, and ensures that the ciphertext is more secure after encryption. The experiments show that the algorithm can encrypt the image effectively and can resist attack on the encrypted image.

## 1. Introduction

With the development of internet technology, a large amount of image information is transmitted in the network, and people can gain the required image information quickly through the internet. However, the internet is an open platform, and images can be easily filched or attacked during their transmission, so finding how to transmit this information securely has become an urgent issue, and also makes digital image encryption one of the research hotspots in the field of encryption. Digital image encryption is an important means to protect image information, and it is of great significance to protect the transmission security of digital images. In recent years, many researchers have mainly studied gray images and color images. Compared with gray images, color images contain rich information, and the requirements for the security of encryption algorithms are more urgent. Due to the non-periodic behavior of chaos, the sensitivity to initial values, and the unpredictability, the chaos theory has been widely used in the image encryption process. Robert Matthews first proposed the idea of encryption based on the chaos theory [1]. Since then, chaos has been widely used in image encryption [2,3,4,5]. So far, many research results exist that are based on chaotic image encryption, such as the Hopfield chaotic neural network [6,7,8], which is a typical dynamic neural network with rich dynamic properties; however, the self-feedback Hopfield network used to generate chaotic phenomena is complex in its structure, computationally intensive with fixed parameters, DNA encryption [9,10,11,12], DNA computation with huge parallelism, and has huge storage and ultra-low power consumption. The compressed sensing (CS) [13,14,15,16] compression feature allows multimedia encryption schemes with a much reduced length of ciphertext, and simple linear measurements make the encryption process very efficient. Therefore, CS-based image encryption schemes have also attracted a lot of attention in recent years. For wavelet transform (WT) [17,18], in digital image encryption schemes, wavelet packet transform (WPT) is often used in the preprocessing stage to decompose the plaintext image, to obtain different image signal components, in order to eliminate some negative factors that are detrimental to the subsequent processing steps and to improve the overall operational efficiency of the image processing scheme. S-box systems [19,20] are used to replace the substitution cipher structure, as a way to ensure the obfuscation performance of the block cipher and the nonlinearity of the generated image, thus improving the performance of the encrypted image. Hyper-chaotic systems [21,22], with multiple initial values and parameters, can greatly improve the key space of encryption algorithms, but the corresponding algorithm complexity increases. Hash functions [23,24,25] play an important role in image encryption systems, and, due to the irreversibility of hash functions, they can resist known plaintext attacks, selective plaintext attacks, and selective ciphertext attacks. In terms of quantum color image encryption schemes [26,27,28,29], quantum-based encrypted images will play an important role as specific and critical quantum information types in the future era of quantum computers. As well as other methods, the literature [30] proposed an encryption, compression and transmission scheme. The scheme is based on a fractional-order chaotic system combined with discrete wavelet transform (DWT) and quadrature phase-shift keying (QPSK) modulation. The cipher performs multiple rounds of digital operations between the vector state of the fractional-order chaotic system and the original image. The transmission process is implemented between a pair of software-defined radio modules, through a QPSK modulation scheme. The literature [31] proposes a two-color image encryption algorithm based on two-dimensional compression perception and wavelet basis, which can simultaneously implement image encryption and compression, and this scheme can effectively reduce the amount of data, and improve the efficiency of transmitting data and distributing keys. However, the complexity is high. The literature [32] proposed a hybrid chaotic fractal system based on a Julia fractal set, and a 3D Lorentzian chaotic system composed of a hybrid chaotic fractal system that generates encrypted images by confusing and diffusing the original image, which has lower computational complexity and provides higher security. The literature [33] proposed a hybrid technique combining Mersenne Twister (MT), deoxyribonucleic acid (DNA), and chaotic dynamical Rossler system (MT-DNA-Chaos), in which the three encryption algorithms not only improve the overall efficiency of data randomization, but the algorithm is more flexible in computation and more secure. The literature [34] proposed a new discrete chaotic encryption algorithm based on DNA coding and SHA-256, but the structure is more complex and there is still room to improve the information entropy after encryption. The literature [35] proposed a novel encryption algorithm based on multiple chaotic mappings, to encrypt the regions of interest (ROI) in images, and this method greatly improves the speed and performance of encryption. The literature [36] introduced an improved Hénon mapping with richer chaotic behavior and better complexity, to improve the security of the color image encryption algorithm, but it uses two-dimensional chaotic mapping with a small key space. Some classical attacks on recently proposed encryption techniques (retrieving the key in a few computations by using a choice plaintext attack and a known plaintext–ciphertext pair) were conducted in the literature [37], to discover the flaws in some of these design structures and suggest some improvements.

Combining the above literature and addressing some of the problems that appear in the literature, this paper proposes a simple color image encryption algorithm that is closely related to the plaintext image and has high security with low complexity. The algorithm introduces the plaintext information and chaotic mapping, which not only improves the key space greatly, but, also, the encrypted result is closely related to the plaintext image, and the key of each image can be guaranteed to be completely different. Moreover, it improves the performance of encrypted image randomness and resistance to differential attack, which ensures that the encrypted ciphertext is more secure. Because the acquisition of the key is closely related to the plaintext, the encryption algorithm proposed in this paper can resist the selected plaintext attack and the known plaintext attack. When the plaintext changes after the dislocation and diffusion operation of the encryption algorithm, the obtained key will also change dynamically, and the correct key will not be obtained. The experiments show that the encryption algorithm proposed in this paper can effectively resist a variety of attacks on encrypted images.

## 2. Basic Theory

This section describes the two chaotic systems, Lorenz system [38] and Arnold mapping [39], required for the encryption algorithm proposed in this paper.

### 2.1. Lorenz System

The Lorenz system, discovered by the American scientist Lorenz in 1963 while studying weather forecasting, is the world’s first dynamical system that exhibits singular attractors and possesses an extremely rich and complex nonlinear dynamical behavior. The dynamic equation of the three-dimensional Lorenz system is shown in Equation (1), as follows:(1)$$\left\{\begin{array}{l} \frac{dx}{dt}=a\left(y-x\right)\\ \frac{dy}{dt}=cx-xz-y\\ \frac{dz}{dt}=xy-bz \end{array}\right.,$$
where *a*, *b*, and *c* are system parameters, taking any value greater than zero, often taking *a* = 10, *b* = 8/3, and *c* as variables. When *c* > 24.74, the system enters a chaotic state. When c = 28, the system enters the optimal chaotic state [40]. In this state, when the initial value is *x*(1) = 0.1, *y*(1) = 1 and *z*(1) = 1, the chaotic attractors generated are shown in Figure 1.

The variation in the three variables *x*, *y*, and *z* with time is nonperiodic and unpredictable, and, as can be observed from the figure, the dynamical orbit is a double helix structure in three dimensions. The helix curves are always confined to a finite space on a plane and they never cut. 

From Figure 1, we can see that the values of *x*(*t*), *y*(*t*), and *z*(*t*) take a good randomness after a certain number of iterations. Therefore, when performing data interception, it is better to avoid the more preceding data, to eliminate the initial state effect. In this paper, the interception starts from the 1000th point to ensure the randomness of the data. 

The chaotic sequences generated by this system have the following advantages: first, the dynamics of the system are more complex than those of the low-dimensional chaotic system, and the resulting numerical sequences have better randomness; second, there are six parameters and initial values of the system, all of which can be used as keys, which greatly enhances the key space; third, there are three chaotic real-valued sequences, which can meet the demand for the direct chaos displacement of the three parameters proposed in this chapter. It can be observed that the set of dynamical equations of the three-dimensional Lorenz system is a ternary system of ordinary differential equations, which requires the solution of this system of ordinary differential equations, and when it is used in digital image encryption, its numerical solution is required, which comes down to the problem of the numerical solution of ordinary differential equations.

**Figure 1 entropy-23-01159-f001:**
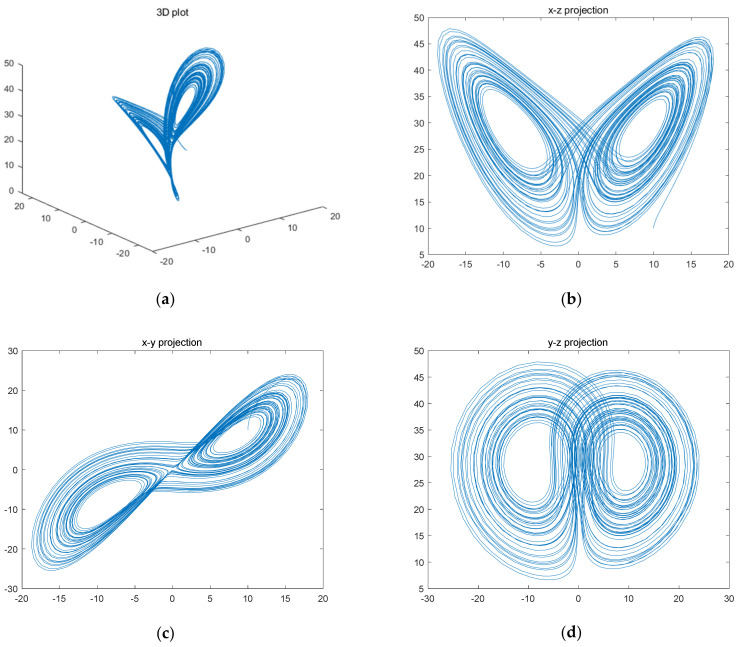
Attractors of Lorentz chaotic system in various dimensions. (**a**) 3D plot; (**b**) x–z; (**c**) x–y; (**d**) y–z.

### 2.2. Arnold Mapping

Arnold mapping, also known as Cat mapping, was proposed by the Russian mathematician Vladimir Igorevich Arnold. It is a chaotic mapping method with repeated folding and stretching transformations in a finite area, and is generally used in multimedia chaotic encryption [41]. 

According to Arnold mapping, the pixels of the original image being transformed are random. However, if the number of iterations is large enough, the original image can eventually be reproduced. This number of iterations is called the Arnold period. The period depends on the size of the image, that is, the Arnold period varies with the size of the image.

Arnold is also considered to be one of the major dislocation algorithms and the algorithm is generated by the following transformation of Equation (2):(2)$$\left[\begin{array}{c} x_{n+1}\\ y_{n+1} \end{array}\right]=\left[\begin{array}{cc} 1 & e\\ d & de+1 \end{array}\right]\left[\begin{array}{c} x_{n}\\ y_{n} \end{array}\right]\mathrm{mod}\left(N\right).$$

Among them, 𝑥_𝑛_,𝑦_𝑛_ denotes the position of the pixel in the grayscale map before the transformation, and 𝑥_𝑛+1_,𝑦_𝑛+1_ denotes the position of the pixel after the transformation, and 𝑑,𝑒 are positive integer parameters, *n* denotes the number of current transformations, *N* is the length or width of the image, and mod(·) is the modulus operation. 

With the forward transformation formula, this algorithm also needs the inverse transformation formula. The inverse transformation formula is shown in Equation (3) below.
(3)$$\left[\begin{array}{c} x_{n+1}\\ y_{n+1} \end{array}\right]=\left[\begin{array}{cc} de+1 & -e\\ -d & 1 \end{array}\right]\left[\begin{array}{c} x_{n}\\ y_{n} \end{array}\right]\mathrm{mod}\left(N\right),$$

The relationship between two matrices transformed into inverse.

The digital image can be seen as a two-dimensional matrix. After the Arnold transformation, the pixel positions of the image will be rearranged, so that the image will be disorganized, thus realizing the scrambling and encryption effect of the image. Arnold transformation has periodicity, that is, after many iterations of Arnold transformation, it will return to the original state. Arnold mapping is generally used for image scrambling. It is mapping from the regular position to a random position, that is, the original strong correlation between the adjacent pixels is replaced by the pixel position, so that the pixels are evenly distributed on the whole image and the correlation between the adjacent pixels is weak. The number of iterations required to repeat the original image is different depending on the size of the image. For a given positive integer N, the period of the transformation is denoted as 𝑇_𝑚_, and when N > 2, the period 𝑇_𝑚_ satisfies 𝑇_𝑚_ ≤ 𝑁^2^ / 2. The periods of Arnold transformations for different orders N 𝑇_𝑚_ are shown in Table 1.

## 3. Encryption and Decryption Algorithm Design

### 3.1. Encryption Scheme

The system proposed in this paper mainly has the following two stages: firstly, the pixel position of the original image is changed by Arnold mapping; secondly, the matrix after Arnold transformation is scrambled by using three chaotic sequences generated by the Lorenz chaotic system; finally, the scrambled RGB components are synthesized into the final encrypted image. Figure 2 is the flow chart of the image encryption algorithm designed in this paper. The specific operation process of the encryption algorithm proposed in this paper is summarized as follows:

Step 1: Extract the RGB component of the color image. The extracted RGB is respectively converted into three 8-bit binary matrices, as follows: $I_{1}$, $I_{2}$, and $I_{3}$.

Step 2: Combine the generated matrix $I_{1}$, $I_{2}$, and $I_{3}$, and normalize it to produce the following three keys associated with the plaintext:$\;a_{1}$, $a_{2}$, and $a_{3}$. These three values are multiplied by the random key q (to enlarge the key space, the value of q can take three different values) as the plaintext-related keys. The multiple keys obtained here are dynamic keys. According to the different images read, the key obtained changes accordingly, that is, the key changes dynamically with the image.

Step 3: set the parameters in the Arnold mapping 𝑑, 𝑒, N = the width of the image (512 in this paper), the 𝑛 number of transformations. 

Step 4: The generated matrices $I_{1}$, $I_{2}$, and $I_{3}$ are transformed by Arnold mapping in step 3, and the RGB matrix components are stretched and folded to change the positions of the pixels in the space, thus destroying the correlation between the adjacent pixel points of the image and generating new matrices $I_{1}^{\prime }$,$\;I_{2}^{\prime },$ and $I_{3}^{\prime }$. 

**Figure 2 entropy-23-01159-f002:**
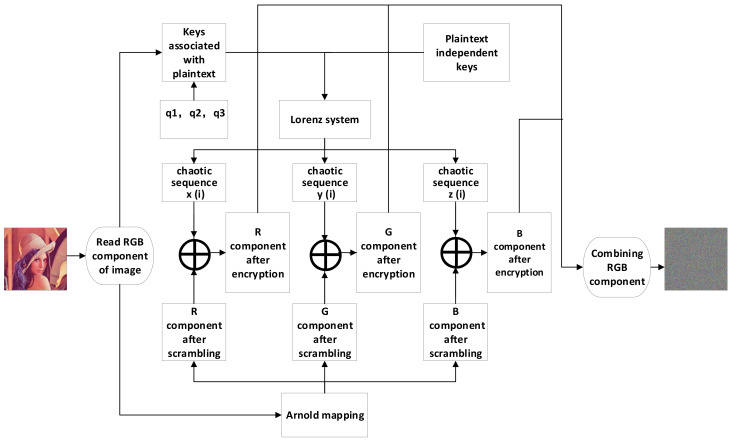
Flow of the designed encryption algorithm.

Step 5: $a_{1}^{\prime }$,$a_{2}^{\prime }$,$a_{3}^{\prime }$,$x_{0}$,$\;y_{0}$,$\;z_{0}$ are the plaintext-independent keys of the chaotic system. $x_{0}$,$\;y_{0}$,$\;z_{0}$ are the initial values of the Lorenz system, respectively, and the parameters of the Lorenz system are generated by Equations (4)–(6). Three chaotic sequences are generated according to the above conditions, and a sequence of length 512 × 512 is selected as the sequence needed to encrypt the image 𝑥(𝑖), 𝑦(𝑖), 𝑧(𝑖).
(4)$$a_{1}^{\prime }+q_{1}a_{1}=a,$$
(5)$$a_{2}^{\prime }+q_{2}a_{2}=b,$$
(6)$$a_{3}^{\prime }+q_{3}a_{3}=c,$$

Step 6: Convert the resulting chaotic sequence 𝑥(𝑖), 𝑦(𝑖), 𝑧(𝑖) elements in the resulting chaotic sequence into a sequence consisting of data between 0 and 255. The formula used here is shown below:(7)$$(a\left(i\right)\times 1000)\mathrm{mod}256,$$

Step 7: the sequences 𝑥(𝑖), 𝑦(𝑖), and 𝑧(𝑖), generated at the end of step 6, are composed from top to bottom and left to right into three 512 × 512 matrices, which are $E_{1},\;E_{2}\;,\;\mathrm{and}\;E_{3}$, respectively, as encryption matrices.

Step 8: Convert the $E_{1},\;E_{2}\;\mathrm{and}\;E_{3}$ data in the encryption matrix into the same type of data as the color image matrix read in step 1.

Step 9: The matrices $E_{1},\;E_{2},\;\mathrm{and}\;E_{3}$ converted in step 8 are XOR calculated with the matrices $I_{1}^{\prime }$,$\;I_{2}^{\prime },$ and $I_{3}^{\prime },$ generated in step 3, and then the encrypted matrix is combined to generate the final encrypted image.

### 3.2. Decryption Scheme 

The decryption algorithm is the inverse of the encryption algorithm, which first reads the ciphertext image, then uses the key transmitted through the secure channel to obtain the chaotic sequence needed for the decryption process, and then XOR the read ciphertext data with the chaotic sequence to obtain the final image after the Arnold-transformed key is inverted by the Arnold transform.

## 4. Simulation Experiments and Performance Analysis

The performance of the proposed image encryption algorithm is analyzed after encrypting and decrypting the Lena color image separately, following the steps given in Section 3. The operating environment of the algorithm is a Windows 10 operating system with a 2.5 GHz Intel CPU I5-4200, 4 GB RAM, and Matlab 2020b.

### 4.1. Encryption and Decryption 

In this section, we use Lena color images (512 × 512) to evaluate the performance of our scheme. During encryption and decryption, our keys are chosen as $\;a_{1}^{\prime }=9.3$, $a_{2}^{\prime }=27.5$, $a_{3}^{\prime }=5/3$, $x_{0}=3.21$, $y_{0}=6.27$, and $z_{0}=1.35$ constants $q_{1}=q_{2}=q_{3}$ = 1. In Arnold transform, 𝑑 = 3, 𝑒 = 9, N = the width of the image (512 in this paper), 𝑛 = 10. Figure 2 shows the encryption and decryption results using our method. From Figure 3, it can be observed that the Lena picture is successfully encrypted and decrypted, and the encrypted image is completely different with respect to the original image.

### 4.2. Histogram Analysis

Histogram analysis is an important security analysis tool in image encryption algorithms, to measure the performance of an encryption scheme in preventing an attacker from accessing the characteristic pixels of an image. Figure 4 shows the histograms of the Lena image (512 × 512), and Figure 5 shows the histograms of the cryptographic image. It can be observed, from the images, that the encryption scheme produces uniform histograms, which implies that our encryption method has good performance against statistical attacks.

### 4.3. Correlation Analysis

The 8-bit nature of digital images results in high correlation between the adjacent pixels. Image encryption schemes can solve this problem by effective pixel dislocation. In this paper, the image is scrambled by Arnold transform and Lorenz chaotic sequence, and the correlation coefficient is used to measure whether the encryption algorithm proposed in this paper has good performance. A good encryption algorithm should generate an encrypted image with low correlation. In this section, we analyze the images before and after encryption by Equation (8).
(8)$$r\left(X,Y\right)=\frac{\mathrm{cov}\left(X,Y\right)}{\sigma \left(X\right)\sigma \left(Y\right)},$$
where *X* and *Y* are two adjacent pixel sequences, cov(X,Y) is the covariance of *X* and *Y*, and $\sigma \left(\cdot \right)$ is the standard deviation. When *Y* is the adjacent horizontal, vertical, and diagonal pixel of each *X*, $r\left(X,Y\right)$ is the horizontal correlation coefficient, vertical correlation coefficient, and diagonal correlation coefficient, respectively. Next,$\;r\left(X,Y\right)$ is used to measure the correlation of two adjacent pixels according to the following criteria: horizontal, vertical, and diagonal. If $r\left(X,Y\right)$ is close to one, there is a high correlation between the adjacent pixels. On the contrary, if $r\left(X,Y\right)$ is close to zero, the correlation between the adjacent pixels is very small.

Five thousand randomly selected pairs of adjacent points were used to calculate the correlation coefficients for each direction (horizontal, vertical, and diagonal). As observed in Figure 6, Figure 7 and Figure 8, the pixels of the plain image (512 × 512) are similar to each other, which depicts that they are highly correlated to each other. However, the adjacent pixels of the encrypted cipher image are not correlated (Figure 9, Figure 10 and Figure 11). Table 2 shows that the correlation coefficient of the original image is close to one, which indicates that the image has a very high correlation for different directions. However, the correlation coefficients for all directions of the encrypted image are close to zero, which indicates that there is almost no relationship between the neighboring pixels. This implies that our encryption method achieves a good encryption effect. In addition, it can be observed from Table 2 that our method has a lower correlation coefficient than that obtained after encrypting the image in the reference, so it shows that the algorithm proposed in this paper has better resistance to statistical attacks.

### 4.4. Information Entropy

In this subsection, color images are used to test the information entropy. The information entropy reflects the degree of confusion of the pixel values of the whole image, and the higher the degree of confusion of the image, the higher the information entropy, and vice versa for the lower the information entropy. The information entropy is calculated by the following formula:(9)$$H=-\sum_{i=0}^{L}p(i)\log_{2}p(i),$$
where L is the gray level of the image and $p\left(i\right)$ is the probability of the occurrence of the gray value i. From Table 3, the information entropy of the original Lena image and the information entropy of the encrypted image can be observed.

It can be observed from Table 3 that the results obtained by our method are close to the theoretical value of eight, which means that the cipher images will be more uniformly distributed and our encryption system is effective for encryption. In addition, according to the data in Table 3, it can be found that the encryption algorithm proposed in this paper has higher information entropy compared with other methods in the literature, which also shows that the encryption algorithm in this paper has a high degree of randomness.

## 5. Security Analysis

In this section, various known analytical methods are used to verify the security of our image encryption method.

### 5.1. Key Space Analysis

For every encryption system, the size of its key space is very important. The larger the key space is, the more it can prevent brute force attacks and increase the time cost of brute force attacks. 

In the algorithm proposed in this paper, the key space consists of the following two parts: one is the key space of the Lorenz system and the other is the key space of Arnold mapping. The keys of the Lorenz system are $q_{1}$,$q_{2}$,$q_{3}$,$a_{1}$,$a_{2}$,$a_{3},a_{1}^{\prime }$,$a_{2}^{\prime }$,$a_{3}^{\prime }$,$x_{0}$,$\;y_{0}$,$\;z_{0}$; the keys of Arnold mapping are d,e,n. The system enters the chaotic state when $a_{1}^{\prime }+q_{1}a_{1}$=10, $a_{2}^{\prime }+q_{2}a_{2}$=8/3,$a_{3}^{\prime }+q_{3}a_{3}>\;24.74\;,$ introduced by the Lorenz system in Section 2.1. When $a_{3}^{\prime }+q_{3}a_{3}$=28, the system enters the optimal chaotic state. So, the key of this chaotic system is chosen as close to the ideal value as possible. Where $q_{1}$,$q_{2}$,$q_{3}$,$a_{1}$,$a_{2}$,$a_{3},a_{1}^{\prime }$,$a_{2}^{\prime }$,$a_{3}^{\prime }$,$x_{0}$,$\;y_{0}$,$\;z_{0}$, the key space of the proposed encryption algorithm in this paper is much larger than $2^{100}$ if calculated with the computer precision of $10^{\left(-15\right)}$, and the key space of the superimposed Arnold mapping and the key space of the Lorenz chaos mapping make the key space much larger than the key space of the Lorenz chaos system, so it is known that the system can effectively prevent brute force attacks.

### 5.2. Key Sensitivity Analysis

Key sensitivity is an important measure of the security of an encryption algorithm. The more sensitive the key is to the value of the initial state of the system, the more secure the algorithm is. Since the generation of chaotic sequences is closely related to each key, any change in the value of any key will lead to the generation of different chaotic sequences, so changing the value of any key can be used as a means to verify the key sensitivity; therefore, in this paper, the key $a_{2}^{\prime }$ is chosen to verify the key sensitivity of this encryption algorithm. The main two aspects of verification are as follows: first, for the same plaintext image, the computer-processable accuracy is 16 bits, as concluded in the key space analysis in Section 5.1, and the smallest computer-processable accuracy is $10^{\left(-14\right)}$ for the key $a_{2}^{\prime }=27.5$, so the other $\delta =10^{\left(-14\right)}$ (the value of $a_{2}^{\prime }$ is converted from 27.5 to 27.5+($10^{\left(-14\right)}$) ) by an order of magnitude, to visually observe the difference between correctly decrypted and incorrectly decrypted encrypted images.

From Figure 12, it can be observed that any small change in the key cannot be decrypted correctly, which also shows that the encryption algorithm is key-sensitive. 

Second, compare the MSE (mean absolute error) values of the original image and the decrypted image to determine the key sensitivity. The mean square error is an IQA (image quality assessment) method, based on signal fidelity (or error signal sensitivity). The MSE is used to calculate the difference between a normal image and a cryptographic image. The degree of similarity between the decrypted image and the original image can be analyzed objectively, and the MAE is calculated as follows:(10)$$\mathrm{MSE}=\frac{1}{M\times N}\sum_{i=1}^{M}\sum_{j=1}^{N}{\left[f(i,j)-g(i,j)\right]}^{2},$$
where f and g are the normal image and its encrypted image, respectively, and M and N are the width and height of the image, respectively. MSE expresses the degree of distortion of an image, according to the statistical characteristics of the signal error, and a larger MSE indicates that the average squared error value of the two images at all pixel positions is larger, i.e., the more the distorted image deviates from the reference image, and the lower its degree of similarity. Conversely, the smaller the MSE value, the smaller the distortion and the higher the degree of similarity. In this paper, the average MSE of the image is used to test the key sensitivity of the image, and the change curve of the average mean square error between the decrypted image and the original image, corresponding to our encryption algorithm when the key $a_{2}^{\prime }$ is varied between $\delta \in \left[-5\times 10^{-9},\;5\times 10^{-9}\right]$, is given in Figure 13. The two arrows on the left side of Figure 13 show the details of the change in the average MSE of the image between –5 × 10^(–9) and –1 × 10^(–9), and the two arrows on the right side of Figure 13 show the details of the change in the average MSE of the image between $-1\times 10^{\left(-14\right)}$ and $1\times 10^{\left(-14\right)}$, and the images show that when the key is changed very slightly, the average MSE of the decrypted image and the original image will change dramatically, and the average MSE of the decrypted image and the original image after the key change is always above 8700.

From Figure 13, we can observe that when the key $a_{2}^{\prime }$ is correct, the MSE value of the decrypted image and the original image is zero, which indicates that the decrypted image is consistent with the original image. When the key $a_{2}^{\prime }$ changes the $\delta =10^{\left(-14\right)}$, the MSE changes greatly, which shows that the encryption algorithm proposed in this paper has very high key sensitivity. In Table 4, by comparing the MSE values of the original and encrypted images in the encryption algorithm proposed in this paper and the previously existing encryption algorithms, it is found that the MSE values of the encryption algorithm proposed in this paper are better. This also indicates that the confidentiality of the encryption algorithm proposed in this paper is higher.

### 5.3. Differential Attacks

To test the differential attack of our method, a number of color images were measured using the number of pixel change range (NPCR) and the uniform average change intensity (UACI), which are defined as follows:(11)$$\left\{{}_{UACI=\frac{1}{M\times N}\sum_{i=1}^{M}\sum_{j=1}^{N}\frac{\left|C1(i,j)-C2(i,j)\right|}{255}}^{NPCR=\frac{1}{M\times N}\sum_{i=1}^{M}\sum_{j=1}^{N}D(i,j)}\right.,$$
where $D\left(i,j\right)$ is a Haveside function, and $D\left(i,j\right)=0$ when $C1\left(i,j\right)=C2\left(i,\text{, }j\right)$ and $D\left(i,j\right)=1$ when $C1\left(i,j\right)\neq C2\left(i,\;j\right)$, according to the ranges provided in the literature [35]. First, we encrypt the original image to obtain a cryptographic image C1. Then, we choose a value and modify it in the original plain image, to obtain another cryptographic image C2. Finally, the NPCR and UACI are calculated by means of Equation (11). The simulation results are shown in Table 5. From Table 5, we can observe that when the image size is 512 × 512, the NPCR value is greater than 0.996 and the UACI values are in the range of 0.33329–0.335541. Therefore, the proposed encryption system can pass a test according to the range proposed in the literature [43], which shows that our image encryption system can effectively resist differential attacks. Moreover, Table 5 shows the NPCR values and UACI values of different image encryption schemes. It can be observed, by Table 5, that these test results are within the acceptable interval, and our encryption method is closer to the theoretical values than other works.

### 5.4. Anti-Noise Capability Analysis

In this subsection, the color image Lena with a size of 512 × 512 is used as the original image, and will be used for encryption by our method. Pepper noise with different noise densities is added to the encrypted image, as shown in Figure 14, and the resulting decrypted image is shown in Figure 15. From these figures, it can be observed that the decrypted image can recover the original image information well when the pepper noise density changes from 0.0001 to 0.01, indicating that the algorithm has some resistance to noise attacks. 

The higher the PSNR, the lower the distortion after compression, and the PSNR is the most common and widely used objective measurement to evaluate the image quality.
(12)$$\mathrm{PSNR}=10\cdot \mathrm{lg}(\frac{L^{2}}{MSE})=10\cdot \mathrm{lg}\left(\frac{L^{2}}{\frac{1}{M\times N}\sum_{i=1}^{M}\sum_{j=1}^{N}{\left(f\left(i,j\right)-g(i,j)\right)}^{2}}\right),$$
Figure 14Encrypted image of different salt and pepper noise density: (**a**) salt and pepper noise density 0.0001; (**b**) salt and pepper noise density 0.001; (**c**) salt and pepper noise density 0.01.
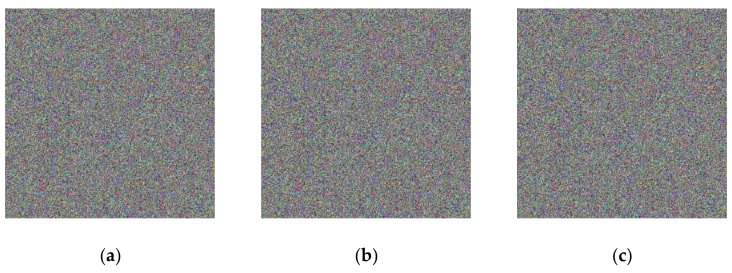

Figure 15Decryption image for different pretzel noise densities;(**a**) salt and pepper noise density 0.0001 decryption image; (**b**) salt and pepper noise density 0.001 decryption image; (**c**) salt and pepper noise density 0.01 decryption image.
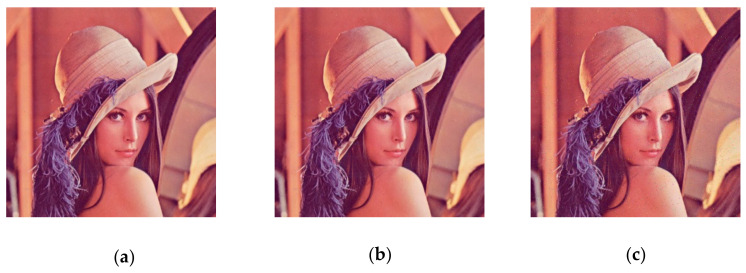


According to Table 6, when the pepper noise density changes from 0.0001 to 0.01, the PSNR of the decrypted image and the original image gradually decreases, but its value is always greater than 27. Generally speaking, it is difficult to observe the distortion when the PSNR of the distorted image is above 35, which indicates the high quality of the image. When the PSNR value of the distorted image is between 28 and 35, the quality of the image will decrease, so the Figure 15, shows that the algorithm has some resistance to noise attack.

### 5.5. Analysis of Shear Resistance

To test the performance of the proposed encryption algorithm against clipping attacks, (a), (b), and (c) in Figure 16 show the encrypted Lena image (512 × 512) after some of the data are clipped off, and (a), (b), and (c) in Figure 17 show the decrypted images corresponding to Figure 16. It can be observed, from the decrypted images, that the cropped encrypted image can also be well recovered by the decryption algorithm, and, therefore, the proposed encryption algorithm has good performance against cropping aggressiveness.

## 6. Conclusions

The algorithm used in this paper introduces the Lorenz system and Arnold mapping; it combines the plaintext key with the image and the key independent of the image as the parameters and initial values of the Lorenz system, so that the key space is greatly improved and the ciphertext is more random and resistant to attack, ensuring that the encrypted ciphertext is more secure. As the key acquisition is related to the plaintext, the key is acquired dynamically, and it is possible to achieve one key per image, which makes the proposed algorithm significantly more secure. Experiments have shown that the algorithm can not only encrypt the image effectively, but can also effectively prevent all kinds of attacks against the encrypted image, such as differential attack, shearing attack, noise attack, etc. Since the plaintext key acquisition in this paper is easier to implement, and the encryption does not decrease due to the reduction in system complexity, the image encryption system proposed in this paper will have better application prospects in the future. However, in the proposed image encryption algorithm, the complexity of the Lorenz system is high and takes up most of the time in encrypting the image information, so, in the future research process, we hope to find a chaotic system with better chaotic characteristics and lower complexity, to replace the Lorenz system, so that the proposed encryption algorithm can be more widely used in subsequent research.

## Figures and Tables

**Figure 3 entropy-23-01159-f003:**
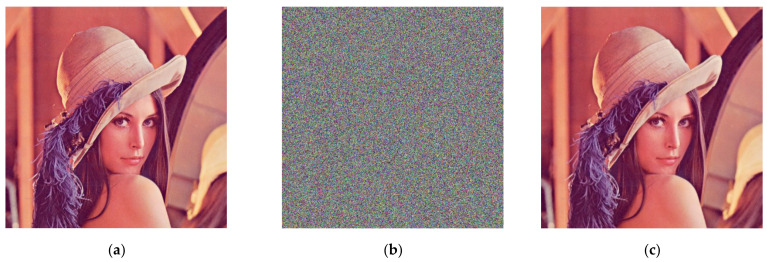
Encrypted and decrypted image of Lena graph. (**a**) The original image; (**b**) encryption image; (**c**) decryption image.

**Figure 4 entropy-23-01159-f004:**
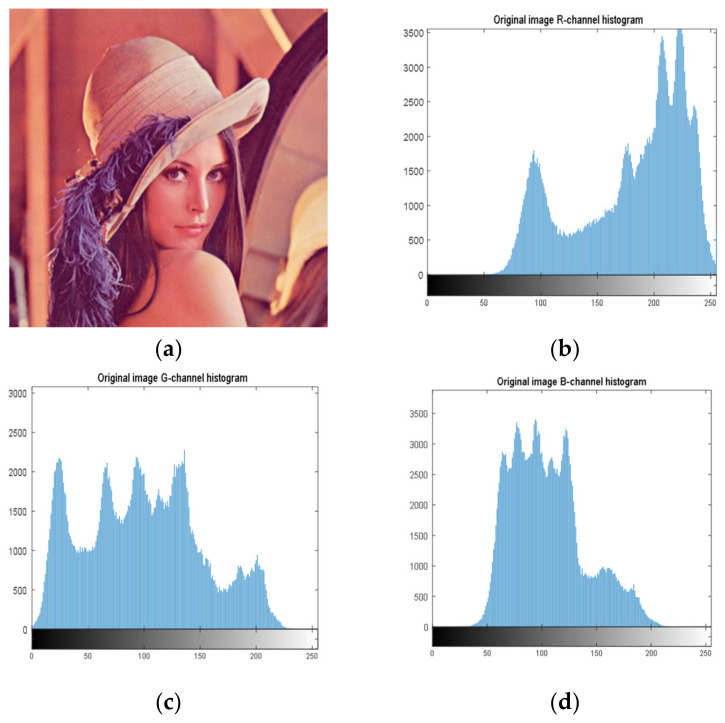
Histogram of components of the original image. (**a**) Original image; (**b**) R-channel histogram; (**c**) G-channel histogram; (**d**) B-channel histogram.

**Figure 5 entropy-23-01159-f005:**
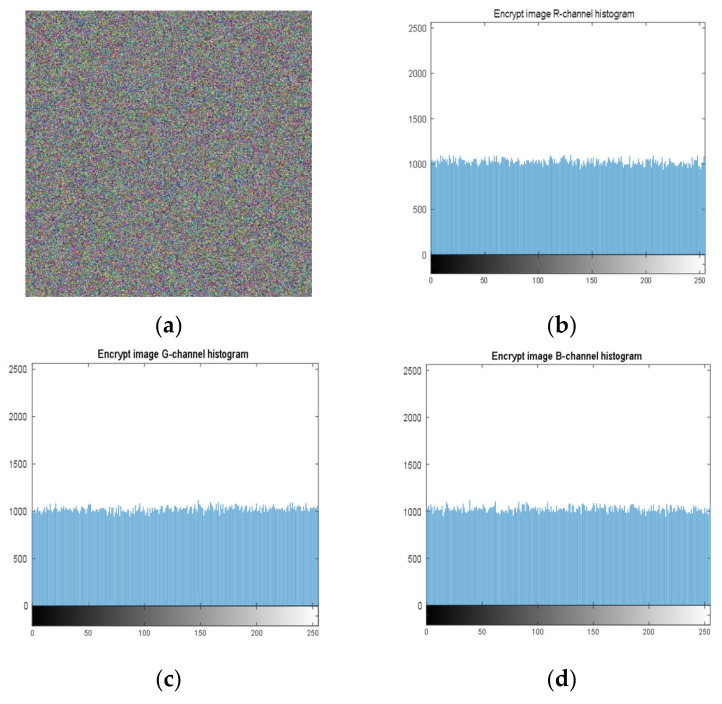
Histogram of each component of an encrypted image. (**a**) Encrypted image; (**b**) R-channel histogram; (**c**) G-channel histogram; (**d**) B-channel histogram.

**Figure 6 entropy-23-01159-f006:**
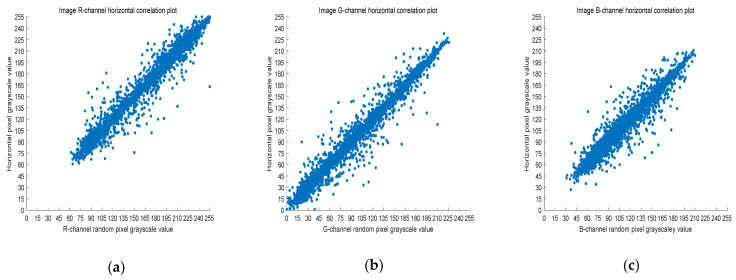
Scatter plot of RGB channels in the horizontal direction of the original image. (**a**) R-channel scatter plot; (**b**) G-channel scatter plot; (**c**) B-channel scatter plot.

**Figure 7 entropy-23-01159-f007:**
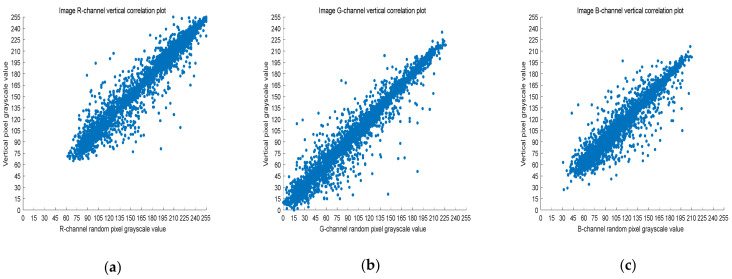
Scatter plot of RGB channels in the vertical direction of the original image. (**a**) R-channel scatter plot; (**b**) G-channel scatter plot; (**c**) B-channel scatter plot.

**Figure 8 entropy-23-01159-f008:**
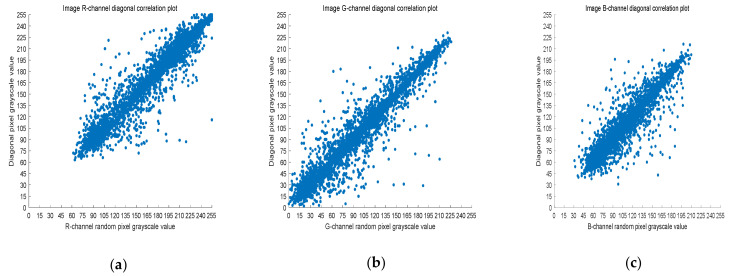
Scatter plot of RGB channels in the diagonal direction of the original image. (**a**) R-channel scatter plot; (**b**) G-channel scatter plot; (**c**) B-channel scatter plot.

**Figure 9 entropy-23-01159-f009:**
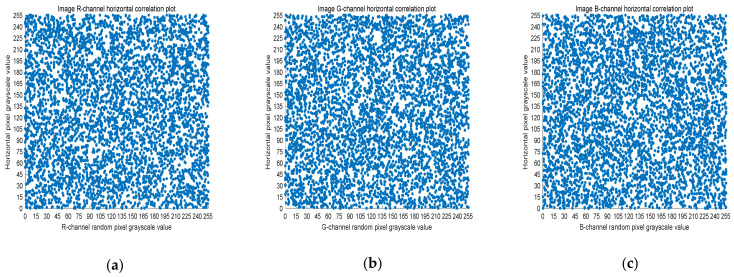
Scatter plot of horizontal RGB component of encrypted image. (**a**) R-channel scatter plot; (**b**) G-channel scatter plot; (**c**) B-channel scatter plot.

**Figure 10 entropy-23-01159-f010:**
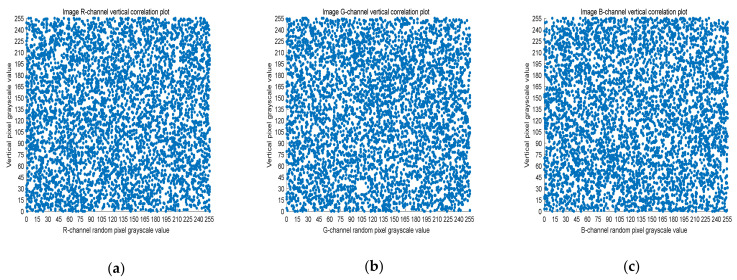
Scatter plot of vertical RGB component of encrypted image. (**a**) R-channel scatter plot; (**b**) G-channel scatter plot; (**c**) B-channel scatter plot.

**Figure 11 entropy-23-01159-f011:**
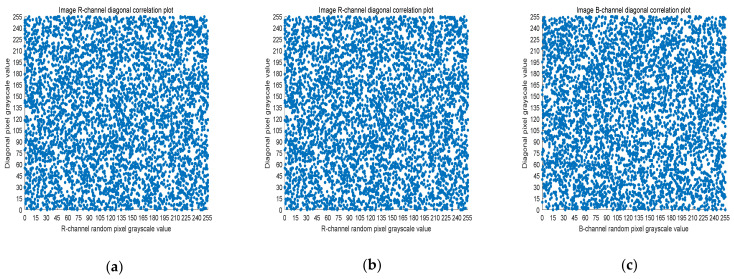
Scatter plot of diagonal RGB component of encrypted image. (**a**) R-channel scatter plot; (**b**) G-channel scatter plot; (**c**) B-channel scatter plot.

**Figure 12 entropy-23-01159-f012:**
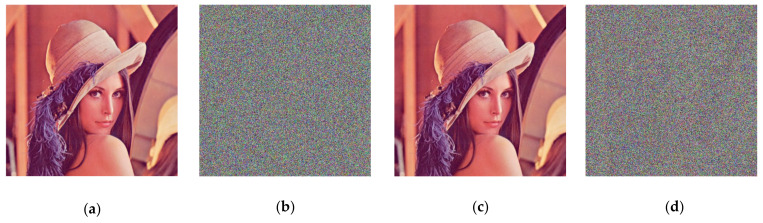
Key sensitivity test image: (**a**) Original picture l; (**b**) encrypted image; (**c**) decrypt image correctly; (**d**) decrypted image after small key change.

**Figure 13 entropy-23-01159-f013:**
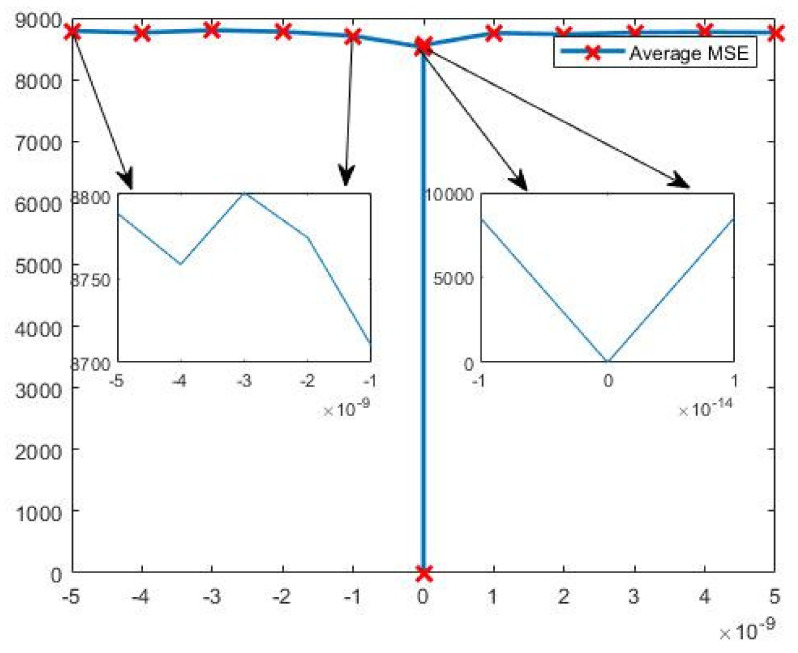
Average MSE corresponding to small change in key.

**Figure 16 entropy-23-01159-f016:**
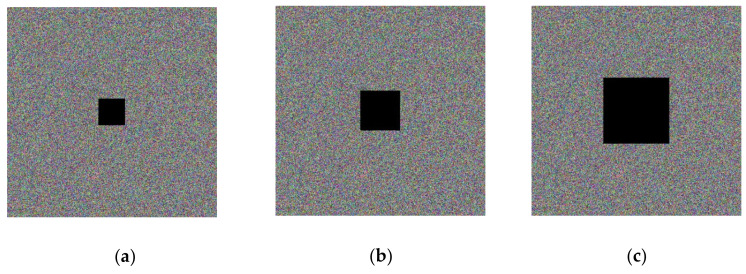
Data loss encryption image.

**Figure 17 entropy-23-01159-f017:**
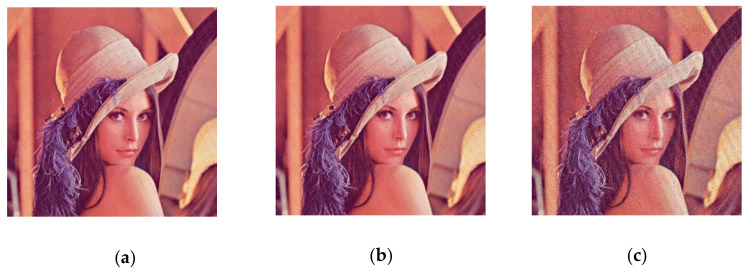
Data loss decryption image.

**Table 1 entropy-23-01159-t001:** Periods of Arnold transformations of different orders N $T_{m}$ [42].

N	2	3	4	5	6	7	8
$T_{m}$	3	4	3	10	12	8	6
N	9	10	11	12	25	50	60
$T_{m}$	12	30	5	12	50	150	60
N	100	120	125	128	256	480	512
$T_{m}$	150	60	250	96	192	240	384

**Table 2 entropy-23-01159-t002:** Correlation of color image components in each direction.

		Horizontal Correlation	Vertical Correlation	Diagonal Correlation
Original image	R	0.97247	0.98656	0.96164
	G	0.97322	0.98675	0.96296
	B	0.94617	0.97208	0.92864
The encryption method proposed in this paper	R	–0.0096211	–0.011037	–0.00084143
	G	0.00096321	–0.0014868	–0.012429
	B	0.0022199	0.0015614	0.0045217
Literature [31]	R	0.0046	0.0046	0.0005
	G	0.0052	0.0058	0.0031
	B	0.0063	0.0084	0.0102
The large aerial images in the literature [33]	RGB average correlation	–0.0014	0.0039	–0.0027

**Table 3 entropy-23-01159-t003:** Information entropy of color images.

	Information Entropy		
Image	R	G	B
original image	7.2682	7.5901	6.9951
Image encrypted in this article	7.9992	7.9993	7.9994
Literature [32]	7.9974	7.9971	7.9975
Literature [34]	7.9973	7.9972	7.9966

**Table 4 entropy-23-01159-t004:** MSE values for different algorithms.

Algorithms	MSE Values
Methodology of this article	8932.0
Literature [35]	7775.0
Literature [33]	9875.5
AES	4600

**Table 5 entropy-23-01159-t005:** Performance against differential attacks.

Lena (512 × 512)	NPCR			UACI		
	R	G	B	R	G	B
Methodology of this article	0.99611	0.99627	0.99616	0.33400	0.33329	0.33483
Tiffany image in the literature [32]	0.9961	0.9961	0.9961	0.3626	0.3626	0.3626
Literature [44]	0.99602	0.99607	0.99601	0.334689	0.334965	0.334155
Literature [45]	0.99640	0.99633	0.99647	0.33488	0.33493	0.33509

**Table 6 entropy-23-01159-t006:** PSNR corresponding to different pretzel noise decryption images.

Image (PSNR)	R	G	B
Pepper noise density 0.0001	48.0455	47.0210	50.5779
Pepper noise density 0.001	37.9455	38.6270	40.1578
Pepper noise density 0.01	27.6638	28.3126	29.5434

## Data Availability

Data is contained within the article.

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
