# Peer review of "Plaintext-Related Dynamic Key Chaotic Image Encryption Algorithm"

_entropy, 2021, doi:10.3390/e23091159_

Round 1

Reviewer 1 Report

The paper report 'Plaintext Related Dynamic Key Chaotic Image Encryption Algorithm'. The system generates dynamic keys, which is the key novelty of this article.  Some statistical analyses are also performed for the evaluation of the algorithm. Moreover, authors have compared their proposed system with some already existing schemes. This paper has significant contributions. Few issues are:

  1. The authors are suggested to add the contribution section or subsection.
  2. There are typo and structural mistakes such as (It can be seen from this table hat the results) hat in Line 284. What is hat? Please proofread the article. Please rewrite line 289 (higher level of randomness more.) Please rewrite Line 287 (proposed in this paper has a 287 higher information entropy.). Check Line 263 (before encryption are closely correlated with each other should be like this: The pixels of plain image are similar to each other which depicts that the they are highly correlated to each other.
  3. Line 107. From the Fig what? Please add the figure number.
  4. Arnold mapping and Lorenz system should be cited properly in their respective section or subsection.
  5. A literature review of some recent chaos-based schemes must be added. https://doi.org/10.3390/rs12111893, https://doi.org/10.3390/e22030274 ,  https://www.mdpi.com/2073-8994/12/3/350.
  6. Literature review of some recent cryptanalysis-based schemes must be added that if the generated key is not dynamic than the system is suspected to several attacks. Please add recent published cryptanalysis based article: https://ieeexplore.ieee.org/document/9492118
  7. In the conclusion part, the authors should highlight that what more improvements can be added in future perspective. How the system can be extended?
  8. Improve the quality of Fig. 2.

Reviewer 2 Report

The paper is well written in a logical manner, comprising a thorough analysis. However, there are some issues needed to be resolved:

1. In the key-space analysis it is needed to explain if all key-values ensure a chaotic steady-state. Maybe some key values lead to a non-chaotic steady-state?

2. In the key sensitivity analysis, the authors have chosen 10e-9 as delta. Did they check what the smallest relevant delta value is? Maybe the method is sensitive for 10e-10? Or for even smaller values?

3. Furthermore, the sensitivity is checked for one parameter only (a2'). What about the sensitivity of other parameters/keys?

4. Does the sensitivity have an impact on key-space? Please elaborate.

5. It is needed to compare the proposed method with similar ones and with standard image encryption methods. What are the advantages/disadvantages? 

Round 2

Reviewer 1 Report

The Paper is revised based on my comments. I recommend this paper for acceptance. The authors have revised the draft paper. 

Reviewer 2 Report

The authors have successfully resolved all issues raised in the previous review.